# Person Re-Identification with RGB–D and RGB–IR Sensors: A Comprehensive Survey

**DOI:** 10.3390/s23031504

**Published:** 2023-01-29

**Authors:** Md Kamal Uddin, Amran Bhuiyan, Fateha Khanam Bappee, Md Matiqul Islam, Mahmudul Hasan

**Affiliations:** 1Interactive Systems Lab, Graduate School of Science and Engineering, Saitama University, Saitama 338-8570, Japan; 2Department of Computer Science and Telecommunication Engineering, Noakhali Science and Technology University, Noakhali 3814, Bangladesh; 3Information Retrieval and Knowledge Management Research Laboratory, York University, Toronto, ON M3J 1P3, Canada; 4Department of Information and Communication Engineering, University of Rajshahi, Rajshahi 6205, Bangladesh; 5Department of Computer Science and Engineering, Comilla University, Kotbari 3506, Bangladesh

**Keywords:** re-identification, video surveillance, multi-modal, cross-modal, RGB–D sensors, RGB–IR sensors

## Abstract

Learning about appearance embedding is of great importance for a variety of different computer-vision applications, which has prompted a surge in person re-identification (Re-ID) papers. The aim of these papers has been to identify an individual over a set of non-overlapping cameras. Despite recent advances in RGB–RGB Re-ID approaches with deep-learning architectures, the approach fails to consistently work well when there are low resolutions in dark conditions. The introduction of different sensors (i.e., RGB–D and infrared (IR)) enables the capture of appearances even in dark conditions. Recently, a lot of research has been dedicated to addressing the issue of finding appearance embedding in dark conditions using different advanced camera sensors. In this paper, we give a comprehensive overview of existing Re-ID approaches that utilize the additional information from different sensor-based methods to address the constraints faced by RGB camera-based person Re-ID systems. Although there are a number of survey papers that consider either the RGB–RGB or Visible-IR scenarios, there are none that consider both RGB–D and RGB–IR. In this paper, we present a detailed taxonomy of the existing approaches along with the existing RGB–D and RGB–IR person Re-ID datasets. Then, we summarize the performance of state-of-the-art methods on several representative RGB–D and RGB–IR datasets. Finally, future directions and current issues are considered for improving the different sensor-based person Re-ID systems.

## 1. Introduction

Person re-identification (Re-ID) has recently gained significant attention among the computer-vision communities and industry. It plays a crucial role in intelligent surveillance systems and has widespread applications for processes including forensic searching; multi-camera tracking, accessing, and controlling; and sports analysis. Person Re-ID is still a challenging task since the videos are recorded by non-overlapping cameras under different environmental conditions, as shown in Figure 1.

In 2006, Gheissari et al. [1] first introduced the concept of person re-identification. This field of computer vision has experienced rapid development, and there has been extensive research performed on this topic. Until now, Re-ID researchers have proposed many outstanding approaches, and most of them are RGB appearance-based Re-ID methods. Over the last few years, Re-ID research has shifted from RGB camera-based Re-ID to different sensor-based (e.g., depth (D) and infrared (IR) cameras) Re-ID as the demand for surveillance systems has increased, especially those that monitor at night in extremely low lighting or dark conditions. Therefore, we aim to provide a quick understanding and overview of the Re-ID researchers who are interested in working with depth and infrared sensors. There are several remarkable person Re-ID surveys that exist in the literature [2,3,4,5,6,7,8,9,10,11,12]. However, these surveys mainly focus on and give insightful direction on RGB camera-based person Re-ID approaches. Recently, Zheng et al. [13] published a survey that focused only on RGB–IR sensor-based Re-ID methods.

**Figure 1 sensors-23-01504-f001:**
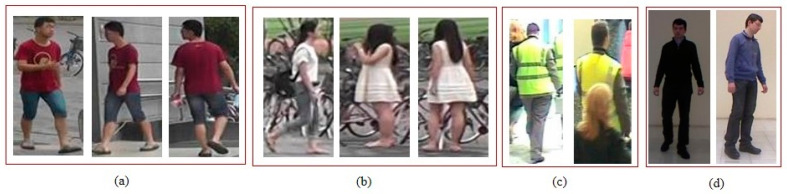
Sample images showing the challenges related to camera variations and environmental conditions in the Re-ID problem. (**a**) shows pose and viewpoint variations; (**b**) background clutter; (**c**) partial occlusion; and (**d**) illumination variations. Images were taken from the public standard datasets IAS-Lab [14], i-LIDS [15] and Market-1501 [16].

Even though the concept of Re-ID was introduced in 2006, Re-ID researchers initially considered only RGB image-based methods; still, these methods are continuing to be used to address challenging issues. In the meantime, modern RGB–D and infrared sensors have been deployed for person Re-ID in 24-h (day and night) surveillance systems. In real-world applications, RGB–D and RGB–IR Re-ID methods are challenging due to significant differences between the two modalities, such as RGB–IR cross-modality matching. In 2010, Jüngling et al. [17] first proposed using an infrared sensor-based image for person re-identification, where they only considered IR–IR video matching rather than the RGB–IR cross-modality one. To address the clothing-change problem for Re-ID, Barbosa et al. first used RGB–D sensors in 2012.

In past years, a lot of Re-ID methods have been proposed to address the constraints of traditional RGB cameras, such as pose [18,19,20], viewpoint variations [21,22], occlusions [23,24,25], illumination changes [26], background clutter [27,28,29], low image resolutions [30,31], and scalability [32]. In some situations, when lighting conditions are extremely poor or dark, it is almost impossible to continue surveillance via RGB cameras as they cannot capture sufficient information in dark environments. In such cases, Re-ID researchers have commonly used modern RGB–D sensors (e.g., the Microsoft Kinect and Intel RealSense depth cameras) and infrared cameras in video surveillance systems. As modern RGB–D sensors avail us with different modalities such as illumination-invariant high-quality depth images, RGB images, and skeleton information simultaneously, as shown in Figure 2, Re-ID researchers have exploited this additional information from sensor-based methods by proposing different approaches to overcome the challenges. Moreover, under normal lighting conditions, they combined RGB and depth information to construct robust features for improving the accuracy of re-identification. While RGB–D cameras can only be used in indoor settings and have distance limitations, infrared cameras have no such limitations and can capture IR images at a wide range of distances in a dark environment.

In this survey, we emphasize the RGB–D and IR sensor-based Re-ID methods in which the query and gallery data come from different domains (i.e., RGB, Depth, Skeleton, and IR). Generally, we divide the RGB–D and IR sensor-based Re-ID methods into three categories: multi-modal, cross-modal, and single-modal person Re-ID (see Figure 3). When RGB data are combined with depth, IR, or skeleton information to improve the Re-ID’s performance, it is called **multi-modal person re-identification**. In many real-world scenarios, when matching RGB with depth or IR modality is important—for example, in a video surveillance system that must recognize individuals in poorly illuminated environments or dark conditions—it is called **cross-modal person re-identification**. When both the query and gallery data come from the same domain (i.e., RGB–RGB, IR–IR, Depth–Depth), it is called a **single-modal Re-ID** problem. All of these modality-aware, Re-ID works consist of conventional methods [14,34,35,36] or deep-learning methods [37,38,39]. Because all the previous surveys [2,3,4,5,6,7,8,9,10,11,12] mainly focus on RGB–RGB single-modality matching, we do not include it in our survey.

In the Re-ID literature, publicly available RGB–D and IR sensor-based datasets [4,14,33,35,39,40,41,42,43,44,45,46] have been recorded for specific purposes because some Re-ID approaches use different domains combinedly to improve the performance of Re-ID [33,34,35,36,37,38,39,47,48] while other approaches use cross-domain methods [49,50,51,52,53] in the Re-ID framework to continue surveillance between daytime visible images and nighttime infrared images. Still, some Re-ID methods consider only IR–IR [17,54] matching using the publicly available datasets.

The main contributions of this work are summarized as follows:First, we categorize the RGB–D and IR sensor-based person re-identification methods according to different modalities. This proposed categorization aims to help new researchers gain a comprehensive and more profound understanding of depth and infrared sensor-based person Re-ID.We summarize the main contributions based on our categorization of person re-identification approaches.We also summarize the performance of state-of-the-art methods on several representative RGB–D and IR datasets with detailed data descriptions.Finally, we give several considerable points for future insights toward improving the different sensor-based person Re-ID systems.

The remaining parts of this survey are organized as follows. In Section 2, we review the multi-modal person re-identification approaches, including RGB–depth, RGB–skeleton, depth–skeleton, RGB–depth–skeleton, and RGB–depth–thermal. Section 3 presents cross-modal person Re-ID approaches, where we review the RGB–depth, RGB–sketch, and RGB–IR cross-modality matching methods and their performances. Existing IR–IR and depth–depth single-modal person Re-ID methods are discussed in Section 4. In Section 5, we give detailed descriptions of the existing RGB–D, RGB–sketch, and RGB–IR person Re-ID datasets. Finally, conclusions and future directions are discussed in Section 6.

## 2. Multi-Modal Person Re-Identification

Most of the existing Re-ID methods focus on matching individuals based on RGB appearance cues. After the arrival of modern RGB–D and infrared sensors, Re-ID researchers took advantage of other modalities, such as depth and IR images, and skeleton information to increase the accuracy of Re-ID. In the past few years, a significant number of works have been proposed to combine RGB appearance cues with depth and/or skeleton information to extract more discriminative features [55,56]. Some works, for example [44], utilized RGB, depth, and thermal information together to improve the performance of Re-ID. Depending on the modalities used by the different Re-ID approaches, we divide the multi-modal person re-identification methods into five categories, depicted in Figure 4. In this section, we overview all five categories of person re-identification methods and their performances on different benchmark datasets.

### 2.1. RGB–Depth Image-Based Person Re-ID

Several recent studies have emphasized improving the performance of Re-ID systems using the additional information from RGB–D sensor-based methods. Some works [37,38] combined RGB appearance features with illumination-invariant anthropometric features from processed depth images. Apart from the above works, some Re-ID methods use raw-depth and RGB images combinedly [40,42,47]. Moreover, in some approaches [48,57], depth frames are used as body segmentation masks, which play an important role in removing cluttered backgrounds from scenes. Most of the above RGB-depth image-based Re-ID methods are considered deep-learning approaches with score-level or feature-level fusion techniques. These approaches can generally be represented using a flowchart, as shown in Figure 5.

In [37], Ren et al. proposed a multi-modal uniform deep-learning (MMUDL) method. Using MMUDL, the authors first processed each depth image into three channels; they computed the value of the depth as the first channel, the angle with the horizontal direction of each point as the value of the second channel, and the height of each point as the third channel. Then, they extracted an anthropometric feature vector by using a CNN and extracted an appearance feature vector from an RGB image by using another CNN. Finally, they combined both feature vectors in a fusion layer with a uniform latent variable that contained three parts: the depth-specific part, the sharable part, and the RGB-specific part. They then computed the distance between the uniform latent variables of two persons for person re-identification. The experiments were conducted using two widely used person Re-ID datasets: RGBD-ID [58] and KinectREID [41]. A cumulative matching characteristic (CMC) was used to evaluate the performance of the Re-ID. The experiments were repeated 10 times and used the average accuracy as the result. The RGBD-ID dataset contains four acquisitions for each individual, and some individuals wore different clothes in different acquisitions. First, the authors removed the individuals who wore different clothes and conducted the experiment without them. It achieved a 100% rank-1 accuracy. When they used the complete RGBD-ID dataset, the experiment achieved a 76.7% rank-1 accuracy. However, when they experimented on the KinectREID dataset, a 97% rank-1 accuracy was achieved. Later, Ren et al. [38] proposed a uniform and variational deep-learning (UVDL) method. Using UVDL, the authors combined depth and appearance features by designing a uniform and variational multi-modal auto-encoder at the top layer of their deep network. This was designed to seek a uniform latent variable by projecting the variable onto a shared space. They achieved a rank-1 accuracy of 99.4% with KinectREID and 76.7% with the RGBD-ID dataset. Though the improvement in rank-1 accuracy for KinectREID was 2.4%, the method failed to improve the results of the RGB-ID dataset.

Some approaches [40,47] use the CNN-based method to fuse local and global features extracted from raw-depth and RGB images. In [40], Lejbolle et al. proposed a multi-modal neural network that was trained to provide fused features using RGB and depth modalities. The authors in [47] improved these methods by proposing a multi-modal attention network (MAT) based on RGB and depth modalities, where they combined a CNN with an attention module to extract local discriminative features and finally fused these local features with globally extracted features. Both approaches tried to overcome some challenges, such as changes in lighting between views, occlusion, and privacy issues, by installing RGB–D cameras on top to capture data from an overhead view. Three RGB–D-based datasets—DPI-T [39], TVPR [43], and OPR [40]—were used for overhead person Re-ID to evaluate the performance of both methods. The results of both methods were presented using CMC curves and their performance was compared using state-of-the-art methods. Lejbolle et al. achieved rank-1 accuracies of 90.36%, 63.83%, and 45.63% for the DPI-T, TVPR, and OPR datasets, respectively, in their first approach. Later, they improved the performance of their Re-ID using a multi-modal attention network. The MAT increased their rank-1 accuracies by 2.01%, 19.15%, and 3.43% for the DPI-T, TVPR, and OPR datasets, respectively.

In [42], the authors proposed a method to fuse RGB and RGB–D image-based ranking scores in a dissimilarity space for person re-identification. They trained two models using either RGB or RGB–D images and then computed dissimilarity scores between the probe and gallery sets using feature embeddings extracted from each model. Finally, both dissimilarity scores were fused in the dissimilarity space to obtain the final recognition accuracy between the probe and gallery sets. Although the computational cost is high, this approach obtained superior accuracies on two publicly available datasets—RGBD-ID [58] and RobotPKU RGBD-ID [33]—and their own dataset, SUCVL RGBD-ID [42]. They achieved rank-1 accuracies of 93.33%, 82.05%, and 87.65% for RobotPKU, RGBD-ID, and SUCVL, respectively. The performance of this method on the RGBD-ID dataset is lower than other datasets because some people wore different clothes in different acquisitions, making it more challenging to recognize people using non-overlapping cameras.

The state-of-the-art approaches described above performed remarkably using RGB and depth data; however, these approaches have some limitations, such as when they follow a feature-level fusion strategy, which may lead to the deep-learning model being overfitted in the fusion of heterogeneous features. Another limitation is that Re-ID systems fail to train good models because depth sensors cannot capture depth frames properly due to the operational range of Microsoft Kinect being between 0.8 to 4 m, which may cause performance degradation.

### 2.2. RGB–Skeleton Image-Based Person Re-ID

Anthropometric measures from depth data are the geometrical features that can describe a person. Anthropometric features are calculated from the joint points of the skeleton body for each person. Although RGB appearance cues are widely used for person Re-ID, these features are not reliable when illumination conditions are very poor. However, color-invariant anthropometric features are reliable under such conditions. Recently, some Re-ID researchers have used color-invariant anthropometric features by fusing them with RGB appearance features to improve the accuracy of re-identification [33,41,59]. A flowchart of the RGB–skeleton information-based Re-ID is shown in Figure 6.

Pala et al. [41] proposed a dissimilarity-based Re-ID framework for fusing anthropometric features, which are extracted from 20 joint points of the skeleton body, with RGB color-appearance features. This dissimilarity-based framework is an alternative to the widely used score-level fusion technique. It used Multiple Component Dissimilarity (MCD) descriptors to reduce the matching time among dissimilarity vectors, which are calculated from different modalities (e.g., body parts). The authors chose three Multiple Part–Multiple Component (MPMC) descriptors, including SDALF [60], eBiCov [61], and MCMimpl [62]. This method was tested on RGBD-ID and their own collected KinectREID datasets, and the results were reported using the CMC curve. They achieved rank-1 recognition rates of 50.37% and 73.85% for the KinectREID and RGBD-ID datasets, respectively.

In [33], the authors proposed an online person re-identification framework where they used appearance and skeleton information provided by RGB–D sensors. In this work, they addressed the drawbacks of offline training approaches due to them not being able to adapt to the changing environment. Instead of offline training, they used an online metric model to update facial information. Then, the authors fused the appearance and skeleton information using a novel feature funnel model (FFM). This approach was validated by them conducting experiments on two publicly available datasets–BIWI RGBD-ID [35] and IAS-Lab [14]—and on their acquired dataset RobotPKU RGBD-ID. Initially, the authors trained the metric model using the RobotPKU dataset and tested it on the IAS-Lab RGBD-ID dataset for both offline and online methods. However, the online method achieved a high performance compared to the offline method. Their online feature funnel model obtained 77.94% and 91.4% rank-1 accuracies for the RobotPKU and BIWI RGBD-ID datasets, respectively.

Distinct from the above methods, Patruno et al. [59] employed Skeleton Standard Postures (SSPs) and color descriptors to re-identify individuals. Three-dimensional skeleton information was used to regularize the pose of every person. Depending on the SSP, a partition grid categorized the samples of the point cloud according to their position. This representation of point clouds provides information about a person’s color appearance. The combination of color and depth information creates an effective signature of the person to increase the performance of re-identification. The effectiveness of this approach was evaluated using three public datasets: BIWI RGBD-ID, KinectREID, and RGBD-ID. The performance of their proposed method was assessed with CMC curves. A rank-1 recognition rate of 61.41% was achieved using this approach on KinectREID, which is an improvement of about 11% compared with Pala et al.’s [41] proposed method. The improvement for RGBD-ID was about 15%, with a rank-1 recognition rate of 89.34% compared with the rate of 73.85% from Pala et al.’s proposed method.

Overall, the state-of-the-art Re-ID approaches described above achieved a moderate accuracy of re-identification using RGB and Skeleton information. One advantage of using skeleton information is that skeletons are invariant under extreme lighting and clothing changes.

### 2.3. Depth–Skeleton Information-Based Person Re-ID

Depth shapes, which are point clouds of individual and skeleton information, are used to re-identify individual people when color-appearance features are unreliable in dark environments. Re-ID researchers have employed depth and skeleton information to tackle extremely low-illumination conditions and the clothing-changes problem. Initially, a skeleton joint-point-estimation algorithm was used to extract soft biometric cues from depth data [58]. To alleviate the problem of individuals being captured in different poses under the distributed cameras in a camera network, some approaches warped the person’s point clouds to a standard pose [14,35]. In some recent works, the depth-shape descriptor of an individual was combined with skeleton-based features to form a complete representation of their human body shape [34,36,63]. The general framework of depth and skeleton information-based Re-ID is shown in Figure 7.

In [58], the authors divided their proposed approach into two distinct phases. In the first phase, they extracted two groups of features: skeleton-based features, which are a combination of the distances between the joint points and the distances between the floor plane and each joint point; and surface-based features, which are the geodesic distances computed between joint pairs. In the second phase, these features were used jointly to re-identify a person by matching a probe set with a gallery setting. The authors evaluated the normalized area under the curve (nAUC) of the CMC to assess the performance of their proposed method. For evaluation, two groups of RGBD-ID datasets (Collaborative and Walking2) were selected. The nAUC was obtained for different features and produced results from 52.8% to 88.1%. Later, the Walking1, Walking2, and Backward groups of the dataset were used to obtain improved nAUC scores.

Munaro et al. [35] proposed a comparison between two techniques, skeleton-based features and global body-shape descriptors, for one-shot person re-identification. First, the authors calculated the few limb lengths and their ratios using the three-dimensional location of the body’s joint points provided by the skeleton estimation algorithm. By exploiting this skeleton information, the method transformed a person’s point clouds into a standard pose in real time to tackle the different poses of an individual. To perform extensive experiments, the authors used the publicly available RGBD-ID dataset and their collected BIWI RGBD-ID datasets, and the CMC and nAUC were used as evaluation parameters. The authors used a skeleton descriptor with four classifiers (Nearest Neighbor, SVM, Generic SVM, and Naïve Bayes), of which Nearest Neighbor achieved the superior results. Their proposed approach obtained a rank-1 recognition rate of 26.6% and an nAUC of 89.7% when the Still group was selected from the BIWI RGBD-ID dataset as a test set, and 21.1% and 86.6%, respectively, for the Walking group. Point-cloud matching outperformed the skeleton-based descriptor with a rank-1 accuracy of 32.5% for the Still group and 22.4% for the Walking group. They evaluated their method using the RGBD-ID dataset and selected different combinations of groups for training and testing. When they selected Walking1 for training and Walking2 for testing, they achieved the best results among the different groups with NN and Generic-SVM classifiers. This method achieved a rank one of 28.6% and an nAUC of 89.9% for NN and 35.7% and 92.8% for Generic SVM, respectively. In [14], Munaro et al. upgraded the previous work by composing 3D models of individual people’s transformed point clouds for re-identification.

In [34], the authors proposed a robust, depth-based person re-identification method where they exploited depth–voxel covariance (DVCov) descriptors and local rotation-invariant depth-shape descriptors to describe pedestrian body shapes. Additionally, they extracted skeleton-based features (SKL) from joint points of the skeleton body and then constructed the re-identification framework by combining the depth shape descriptors and skeleton-based features. The proposed framework was evaluated on three re-identification datasets: RGBD-ID, BIWI RGBD-ID, and IAS-Lab RGBD-ID. It achieved a high performance with the RGBD-ID dataset, where the rank-1 recognition rate was 67.64% for single shots and 71.74% for multi-shots, and outperformed all depth- and skeleton-based methods. When it was used with the BIWI and IAS-Lab datasets, its performance decreased because of the different viewing angles that made the analysis more challenging than it was with the RGBD-ID dataset.

In [63], Imani et al. proposed a short-term Re-ID method where they modeled depth images as complex networks and introduced two novel features, named the histogram of the edge weight (HEW) and the histogram of the node strength (HNS), on the networks. However, the authors also extracted HNSs from single frames and multi-frames and defined them as histograms of special node strength (HSNS) and histograms of temporal node strength (HTNS), respectively. Finally, these features were combined with skeleton features using score-level fusion. They selected two short-term person re-identification datasets, RGBD-ID and KinectREID, to evaluate the performance of their proposed approach. The proposed approach achieved performances at rank one of 58.35% and 64.43% for single shots and multi-shots, respectively, for the KinectREID dataset. However, with the RGBD-ID dataset, it achieved 62.43% and 72.35% at rank one with single shots and multi-shots, respectively.

Later, Imani et al. [36] proposed a person re-identification method using three local pattern descriptors (local binary patterns (LBP), local derivative patterns (LDP), and local tetra patterns (LTrP)) and anthropometric measures. This approach fused the histograms of LBP, LDP, and LTrP with the anthropometric measures using Kinect sensors. As their proposed approach is for short-term person Re-ID, they selected two short-term re-identification datasets, RGBD-ID and KinectREID, for their experiments. The RGBD-ID dataset was categorized into four groups—Backward, Walking1, Walking2, and Collaborative—according to their frontal and rear views. The people who changed their clothes were discarded from the experiments and divided into two smaller databases, Walking2–Backward and Walking1–Collaborative. The fusion of SGLTrP3 (i.e., third-order LTrP with Gabor features) with anthropometric measures achieved rank-1 accuracies of 76.58% and 72.58% for the Walking1–Collaborative and Walking2–Backward groups, respectively. The Walking1–Collaborative group achieved a higher accuracy than Walking2–Backward because it has a frontal view while the latter group has a rear view. The recognition rate of the KinectREID dataset at rank one of the SGLTrP3 was 66.08%.

Although the performance of depth and skeleton information-based Re-ID is lower than that of color-appearance-based Re-ID, it is very effective where RGB appearance-based Re-ID methods tend to fail, i.e., in dark conditions or when clothes are changed.

### 2.4. RGB-, Depth-, and Skeleton-Based Person Re-ID

Because RGB–D sensors can capture RGB, depth, and skeleton information simultaneously, Imani et al. [55] used depth and RGB feature descriptors along with skeleton information. They proposed a local pattern descriptor (LVP—local vector pattern) to extract features from different regions of depth and RGB modalities, where depth and RGB images were divided into three regions: the head, the torso, and the legs. Skeleton features, which are particularly Euclidean distances, are computed from 20 joint points of a skeleton. Finally, features extracted from the different modalities were combined by double and triple combinations using score-level fusion as shown in Figure 8. However, the performance of the triple combination (i.e., RGB, depth, and skeleton) did not exceed the double combination (i.e., depth and skeleton, RGB and depth, or RGB and skeleton).

They performed experiments on two publicly available datasets, KinectREID and RGBD-ID, and investigated different combinations of the modalities. A combination of skeleton and RGB modalities achieved the highest level of accuracy, with a rank-1 recognition accuracy of 75.83% with the KinectREID dataset. The authors split the RGBD-ID dataset into two smaller datasets, named RGBD-ID1 and RGBD-ID2, by discarding the people who changed their clothes in the different groups. RGBD-ID1 consists of the Walking1–Collaborative group of 59 people and RGBD-ID2 consists of the Walking2–Backward group of 72 people. The recognition performance of RGBD-ID1 exceeded the RGBD-ID2 dataset for the combination of depth and skeleton modalities because the Walking1 and Collaborative groups are in frontal view. The recognition accuracies were 85.5% and 80.5% for the RGBD-ID1 and RGBD-ID2 datasets, respectively.

The main challenge of the tri-modal method (i.e., RGB, depth, and skeleton) is how to fuse the features of the three modalities to gain a superior accuracy. A combination of RGB and skeleton features achieved a higher level of accuracy than the tri-modal method, which is a drawback of this approach.

### 2.5. RGB, Depth, and Thermal Image-Based Person Re-ID

Mogelmose et al. [44] proposed a novel approach that integrated RGB, depth, and thermal data in a person Re-ID system. The authors used color histograms for RGB data from different regions of the body, and soft biometric cues were extracted from the depth data. Because thermal images cannot contain color information, they extracted local structural information using SURF [64]. The features extracted from the three modalities were fused in a joined classifier as shown in Figure 9. The proposed tri-modal system was evaluated with their own recorded RGB–D–T dataset using a CMC curve, and it gained an 82% rank-1 recognition accuracy.

Summarized information about the above studies is presented in Table 1.

## 3. Cross-Modal Person Re-Identification

Single-modal person Re-ID matches probe samples with gallery samples, and all samples are taken from the same modality (i.e., RGB–RGB or IR–IR matching) [65]. Unlike single-modal Re-ID, cross-modal Re-ID aims to match the probe sample taken from one modality against a gallery set from another modality, such as RGB–IR, RGB–depth, or sketch–RGB images [13,52,66], as shown in Figure 10.

Cross-modal Re-ID is very important for nighttime surveillance systems because the RGB camera fails to capture color information at night. On the contrary, depth and IR cameras can capture depth and infrared images, respectively. Re-ID researchers have employed both depth (i.e., RGB–D) and IR cameras for nighttime surveillance. However, RGB–D cameras (e.g., Kinect and Intel RealSense depth cameras) are independent of visible light, have distance limitations, and can only be used for indoor applications. In contrast, modern surveillance cameras can automatically switch from RGB to IR in dark environments. Therefore, both cameras (i.e., depth and IR cameras) can be employed in cross-modal Re-ID for 24-h surveillance systems. Considering the availability of different modalities, cross-modal person Re-ID can be divided into three categories: sketch–RGB cross-modal Re-ID, RGB–depth cross-modal Re-ID, and RGB–IR cross-modal Re-ID. In this section, we overview all the categories and data discrepancies among the different modalities used in cross-modal Re-ID and briefly describe the state-of-the-art performance of the proposed methods.

### 3.1. RGB–Sketch Cross-Modal Person Re-ID

To give a real-world scenario, sometimes it is required to investigate a crime scene when a query person is unavailable in practical forensic settings. Pang et al. [66] proposed a cross-modal person Re-ID method using a sketch of a target person. A sketch of a person carries low-level texture information without color cues. The authors employed a cross-domain adversarial feature learning technique that jointly learns identity- and domain-invariant features. This method significantly reduces the domain gap between sketch and RGB modalities. They performed experiments on their proposed sketch-photo Re-ID dataset and evaluated the results using CMC curves. Although the authors did not specify which domain was selected as the query set and which as the gallery set, it might have been sketch as the query and RGB as the gallery, and vice versa. The proposed method obtained a 34.0% rank-1 accuracy.

### 3.2. RGB–Depth Cross-Modal Person Re-ID

RGB–depth cross-modal Re-ID matches a person of interest across RGB and depth modalities. Recently, a few works [49,50,51,52,53] investigated how to mitigate the domain gap between RGB and depth modalities. As the nature of the data for both modalities is heterogeneous, Re-ID researchers have proposed their own methods to tackle the challenges of the process, such as data discrepancy, and have utilized conventional [49,50] and deep-learning methods [51,52,53] to identify individuals across the two modalities. The general architecture of RGB–depth cross-modal Re-ID methods is shown in Figure 11.

In [49], the authors proposed a heterogeneous camera network to re-identify a person across RGB and depth modalities. To maintain a correlation between these two modalities, a dictionary-based learning algorithm was proposed to transform the edge gradient features (HOG and SILTP) extracted from RGB images and Eigen-depth features extracted from depth images into sparse code that shares a common space. The proposed method was evaluated with two datasets: RGBD-ID and BIWI RGBD-ID. Experiments were carried out for two cases, one which used RGB image-based features for the gallery and depth-based features for the probe, and another which was reversed. For the RGBD-ID dataset, a rank-1 recognition rate was obtained at 11.05% when they set depth-based features as the gallery and RGB-based features (HOG) as the probe. Similarly, they obtained an 11.84% rank-1 recognition for the BIWI RGBD-ID dataset. However, the rank -one recognition rate was highest (12.11%) when RGB-based features were extracted using SILTP and set as the probe, and depth-based features were set as the gallery. Uddin et al. [50] also used edge gradient features (HOG, SILTP, and LBP) as local shape descriptors, but in a slightly different manner than the above work [49]. The authors extracted edge gradient features from both modalities (RGB and depth) and used PCA- and LDA-based metric learning approaches to obtain a higher accuracy of Re-ID. They performed experiments on the BIWI RGBD-ID and IAS-Lab RGBD-ID datasets and received higher accuracies using HOG-based features for both datasets. The rank-1 recognition rates were 41.43% and 38.93% for the BIWI and IAS-Lab datasets, respectively, when they considered RGB as the gallery and depth as the probe.

Recently, Hafner et al. [51,52] proposed a deep transfer learning technique where a cross-modal distillation network was employed to transfer knowledge from RGB modalities to depth modalities or vice versa to solve cross-modal Re-ID between RGB and depth. The authors extended the work of [51] by changing embedding size as a hyperparameter for the optimization of Re-ID to achieve a superior accuracy [52]. They evaluated their method using two public datasets, BIWI RGBD-ID and RobotPKU, and obtained a 42.8 ± 3.9 % rank-1 recognition for BIWI with RGB as the gallery and depth as the query. However, for the RobotPKU dataset, they received a higher accuracy (25.3 ± 2.0%) when RGB was set as the query and depth as the gallery. Although this approach obtained outstanding performance, the cross-modal distillation network is a two-stage learning procedure and cannot be trained in an end-to-end manner. Additionally, this network extracts only global features from all images of individual pedestrians, which affects the viewpoint variations of the pedestrians. To solve this issue, Wu et al. [53] proposed an end-to-end heterogeneous restraint network (HRN) to perform RGB–D cross-modal person Re-ID tasks. HRN captures common spatial relationships between RGB and depth modalities by using a dual-modal local branch and reduces the domain gap between the two modalities. This approach was tested on two datasets, BIWI RGBD-ID and RobotPKU, with one setting including RGB as the gallery and depth as the query and another setting with the opposite configuration. When the authors used depth as the query and RGB as the gallery for the BIWI dataset, they achieved a higher rank-1 accuracy (47.1%) compared with the opposite setting (43.9%). However, this approach obtained an only 25.7% rank-1 recognition accuracy for the RobotPKU dataset when they used depth as the query and RGB as the gallery.

### 3.3. RGB–IR Cross-Modal Person Re-ID

RGB–IR-based Person Re-ID is the most widely studied cross-modal setting over all the other alternatives, thanks to the introduction of the SYSU-MM01 dataset [45], which initiates the path of an RGB–IR-based cross-modal Re-ID scenario. Following the survey paper in [13], state-of-the-art Re-ID approaches using RGB–IR-based cross-modal methods can be divided into two categories: non-generative- [67,68,69,70,71,72,73,74,75,76,77,78,79,80,81] and generative-based approaches. The former one relies on traditional feature representation [67,68,69,70,71,72,73,74,75,76,77,78,79,80,81] and metric learning approaches to maximize the similarities between two images with the same identity and minimize the similarities between two images with different identities, while the latter one depends on the unification of images from different modalities to minimize the data distribution gap between two different modalities.

Non-Generative Cross-Modal Re-ID: Non-generative cross-modal Re-ID approaches aim to find discriminative feature embedding by either deploying feature learning approaches [67,68,69,70,71,72,73,74,75,76,77,78,79,80,81,82,83,84,85,86,87] or utilizing metric learning approaches. Most of the existing feature learning approaches focus on extracting global features by employing: one individual branch each for visible and infrared images [67]; a cross-modality shared-specific feature transfer algorithm [68]; consistency at the feature and classifier levels [69]; or attention mechanisms [70,71]. Some global features that learning Re-ID approaches focus on are the disentanglement of ID-discriminable and ID-ambiguous cross-modality feature subspaces [72] and the disentanglement of spectrum information to maximize invariant ID information while minimizing the influence of spectrum information [73]. Contrarily, local feature learning-based Re-ID approaches [74,75,76,77,78,79,80,81] rely on fine-grained person recognition by either dividing global features into part-based feature representations [74] or deploying a body partition model [75] to automatically detect and distinguish effective component representations. There are some Re-ID approaches [76,77,78,79,80,81] that utilize both global and local features to design powerful feature descriptors for RGB–IR-based cross-modal Re-ID settings.

The aim of metric learning-based Re-ID approaches [82,83,84,85,86] is to direct feature representations to fulfill certain objective functions for better recognition. Owing to the metric learning approach, a two-stream network architecture was proposed in [82], in which contrastive loss is utilized to bridge the gap between two modalities and enhance the modality invariance of learned representations. Following the success of utilizing triplet loss for classical Re-ID, Liu et al. [83,85] reformulated triplet loss with a center-to-center mode rather than instance-to-instance. Ye et al. [84] proposed a triplet loss that uses the angles between instances as a distance measure for optimizing the network. Similarly, in [86], a feature space is mapped onto an angular space and utilized for triplet loss and center loss for network optimization. In [87], a modified version called quadruplet loss is proposed, which combines triple loss with classification loss to optimize the proposed hyperspherical manifold-embedded network.

Generative Cross-Modal Re-ID: The recent success of generative adversarial networks (GANs) for generating fake images has prompted researchers to focus on mutual translation between the two modalities. In contrast to direct modality translation, a few approaches disentangled ID-discriminative and ID-excluding factors and then generated image pairs to extract highly discriminative features. In direct modality translation, Zhong et al. [88] proposed a colorization-based Siamese generative adversarial network that bridged the gap between modalities by retaining the identities of the colored infrared images. Regarding the matching from visible to infrared, in [88], fake infrared images generated by GAN matched with gallery images to minimize the discrepancies between the modalities. Because of the claim that GAN-generated fake images introduce plenty of noise, fake images generated by GAN were replaced with grayscale images that had three channels in [89]. Projecting both visible and infrared modalities into a common consistent space seemed to be successful in [90,91,92,93,94], reducing the effects of bias caused by individual modality. Most of these approaches [90,91,92,93,94] first generated visible–infrared image pairs through disentanglement and then mapped them onto a unified space to reduce their dual-level discrepancies.

Apart from modality-translation-based Re-ID approaches, there are a few attempts [95,96,97,98] that introduce a third modality to reduce the modality discrepancy. The idea of using a third modality was proposed by Li et al. [95], who introduced an “X” modality as a middle modality to eliminate cross-modal discrepancies. Following the same pipeline, in [96,97], real images from both modalities were combined with ground-truth labels to generate third-modal images, which help to reduce modality-related biases. In [98], a non-linear middle modality generator was proposed that effectively projects images from both modalities onto a unified space to generate an additional modality to reduce the modality discrepancies.

A thorough review of state-of-the-art cross-modal Re-ID approaches suggests that the recognition accuracies (i.e., rank-1 and mAP) of all these approaches are still below average when compared with other Re-ID scenarios. For instance, the rank-1 accuracy of Pang et al.’s [66] approach with an RGB–sketch cross-modal person Re-ID setting is only 34%, whereas the range of rank-1 accuracies for most state-of-the-art approaches with RGB–depth cross-modal Re-ID settings is between 40% and 50%. Compared with the two cross-modal settings described above, the range of rank-1 accuracies for RGB–IR cross-modal Re-ID is higher, mainly due to fewer inter-modal data discrepancies between RGB and RGB–IR modalities. From this discussion, it can be deduced that there is ample room for improving the accuracy of Re-ID, and it is still an active area of research.

## 4. Single-Modal Person Re-ID

In this survey, we include only IR–IR and depth–depth single-modal person Re-ID because previous surveys did not include these types of single-modal Re-ID methods [2,3,4,5,6,7,8,9,10,11,12]. They mainly focused on RGB–RGB single-modal person Re-ID methods. IR–IR and depth–depth modality matching is a very challenging task because IR and depth images have no color appearance cues, even texture information.

### 4.1. IR–IR Single-Modal Person Re-ID

To the best of our knowledge, only two Re-ID works [17,54] have been proposed with IR–IR matching. In [17], the authors introduced a person re-identification approach utilizing local features in infrared image sequences. Local features are visual words that are used to generate a person’s signature. Then, these features were used for person re-identification. They evaluated their approach using a subset of the CASIA infrared dataset [99] and received correct classification rates of 81% for single frames and 95% for sequences of frames.

In [54], Coşar et al. proposed a re-identification method for service robots using thermal images, in which the authors performed entropy-based sampling during training to build thermal dictionaries of humans and then converted the sequence of dictionary elements into a symbolic form. This new representation of symbols was used as a feature to train SVM classifiers. They recorded a thermal dataset to perform experiments for person re-identification. The dataset was recorded in an indoor environment for different situations, including human motion, poses, and occlusion, using a thermal camera (Optris PI-450) which was mounted on a Kampai robot. Finally, an experiment was performed on the dataset and obtained a 57.6% overall re-identification accuracy for their proposed method.

### 4.2. Depth–Depth Single-Modal Person Re-ID

In Section 2.3, we summarized the works [34,35,36,58,63] that combined depth shape and skeleton joint-point information to re-identify a person. Among them, Wu et al. [34] additionally investigated depth–depth single-modal person Re-ID on three benchmark datasets, RGBD-ID, BIWI RGBD-ID, and IAS Lab, where they used DVCov (depth–voxel covariance) and ED (Eigen-depth) features. With the RGB–D dataset, they obtained 61.49% and 44.67% rank-1 accuracies for DVCov and ED features, respectively, when one image per person was randomly selected as the gallery (single-shot). Similarly, rank-1 accuracies of 16.32% and 28.98% with the BIWI RGBD-ID dataset were achieved, and 27.95% and 32.09% with the IAS Lab dataset. In [52], Hafner et al. proposed the cross-modal person Re-ID method discussed in Section 3.2. The authors also investigated depth–depth single-modal Re-ID and evaluated its performance with BIWI and RobotPKU datasets. In this case, they used ResNet50 as an optimizer with both triplet loss and softmax loss and obtained a higher performance with softmax loss. With BIWI, it achieved an approximately 59.8% rank-1 recognition accuracy. With RobotPKU, it achieved an approximately 44.5% rank-1 recognition accuracy.

To evaluate all the proposed Re-ID approaches, it is of utmost importance to have proper Re-ID datasets with proper experimental settings. Consequently, contributing a new dataset is considered one of the main contributions in the literature. Thus, considering the importance of Re-ID datasets, the following section (i.e., Section 5) discusses the most important multi-modal Re-ID datasets that are used by the Re-ID community to validate their approaches.

## 5. Datasets

There are several surveys [2,3,4,5,6,7,8,9,10,11,12] that mainly focus on RGB Re-ID dataset description and summarization. There is no proper guidance for existing benchmark RGB–D, IR, thermal, and sketch Re-ID datasets. This section summarizes the frequently used RGB–D, IR, thermal, and sketch person Re-ID datasets. The datasets are divided into five categories depending on widely used surveillance applications and modalities.

### 5.1. Multi-Modal RGB–D Datasets

As RGB cameras cannot capture the color cues of individuals in poor illumination conditions, Re-ID researchers have employed RGB–D sensors (Microsoft Kinect and Intel RealSense depth cameras) to capture images of individuals in dark conditions. RGB, depth, and skeleton modalities can be obtained simultaneously using Microsoft Kinect SDK [35], as shown in Figure 2. By combining these modalities, Re-ID researchers have proposed and evaluated their methods using benchmark datasets, such as RGBD-ID [58], KinectREID [41], RobotPKU RGBD-ID [33], SUCVL RGBD-ID [42], BIWI RGBD-ID [35] and IAS-Lab [14]. These datasets were recorded with different variations of poses, including the frontal, rear, and side views of individuals as shown in Figure 12. The datasets are summarized in Table 2.

### 5.2. Top-View RGB–D Datasets

Unlike the multi-modal datasets described above, these were captured from a top view rather than horizontally to reduce occlusion and privacy [43], as shown in Figure 13. Three RGB–D datasets have been proposed for top-view person re-identification: Top-View Person Re-Identification (TVPR) [43], Depth-Based Person Identification from Top (DPI-T) [39], and Overhead Person Re-Identification (OPR) [40].

TVPR: This dataset comprises 23 video recordings of 100 people containing RGB and depth information. Recordings were made in an indoor hall, where people were passing under the camera installed on the ceiling. Most of the methods split the dataset into a training set, which consists of all people walking from the left to the right directions, and a testing set, which consists of the same persons walking in the opposite direction [40,43,47].

DPI-T: This dataset was recorded with an RGB–D camera in an indoor hall and consists of 12 persons who appear in a total of 25 videos across several days. On average, individuals wore five different sets of clothing during the recording times.

OPR: This dataset was collected using a depth camera which was placed in the ceiling at a university canteen to capture images of individuals when entering and leaving the canteen. This dataset consists of 78,742 frames and 64 different persons.

### 5.3. RGB–D and Thermal Dataset

An RGB–D and thermal dataset named RGB-D-T [44] contains data from three modalities, RGB, depth, and thermal (see in Figure 14), of 35 people who passed by the sensors twice for 70 passes in total. To capture all three modalities at one time, a Microsoft Kinect camera was used to capture RGB and depth data, and a thermal camera was mounted over the Kinect’s RGB camera lens.

### 5.4. RGB–Sketch Re-ID Dataset

Unlike the above datasets, this dataset contains whole-body sketches and RGB images of individuals (see Figure 15), where sketches are used to match against RGB images. Sketch–RGB cross-domain matching is a very challenging task due to the large gap between sketch and RGB data. Pang et al. [66] collected a sketch Re-ID dataset with 200 people, where each individual has one sketch and two RGB images from the different associated cameras.

### 5.5. RGB–IR Cross-Modal Datasets

RGB–infrared Re-ID aims to match daytime RGB images and nighttime infrared images. This is a cross-modality matching because the two modalities have significant gaps according to the imaging principle (i.e., wavelength range) and nature of the data. In recent years, Re-ID researchers have focused on cross-modal matching and have proposed several methods. Almost all methods [67,68,69,70,71,72,73,74,75,76,77,78,79,80,81,82,83,84,85,86,87,88,89,90,91,92,93,94,95,96,97,98] of RGB–IR cross-modal Re-IDs used two benchmark datasets: SYSU-MM01 [45] and RegDB [46]. To perform experiments in their proposed methods, they used two RGB–IR benchmark datasets: SYSU-MM01 [45] and RegDB [46].

SYSU-MM01: This is the first public benchmark RGB–IR dataset, proposed in 2017. It was collected using six cameras: four RGB cameras and two IR cameras. IR cameras were put in two dark places (Camera 3 was in the indoor room and Camera 6 in the outdoor passage) to capture IR images while RGB cameras (Camera 1, Camera 2, Camera 4, and Camera 5) were installed in four different indoor and outdoor places to capture RGB images, as shown in Figure 16. There are, in total, 491 persons, 287,628 RGB images, and 15,792 infrared images with different poses and viewpoints. Each person was captured by at least two cameras.

RegDB: RegDB is a small-scale dataset that contains 412 persons captured by a dual camera system (one RGB and one thermal camera). Among the 412 persons, there are 254 females and 158 males. Each person appears in 10 RGB images and 10 corresponding thermal images. There are a lot of viewpoint variations in this dataset because 156 individuals were captured from the frontal view, while the remaining 256 were captured from the rear view.

## 6. Conclusions and Future Directions

Although visible (RGB) camera-based person Re-ID has obtained popularity in the last decade, increasing application requirements in the nighttime have caused depth and infrared sensor-based Re-ID to attract some researchers’ attention in the last few years. This paper presents a comprehensive survey of person Re-ID with RGB–D and RGB–IR sensors. First, we divide existing RGB–D and RGB–IR sensor-based works into three different categories, multi-modal, cross-modal, and single-modal, and then these are subcategorized according to the modalities used in the different methods of Re-ID. This categorization will help new researchers who want a quick understanding of specific modality-based Re-ID approaches. We also summarize the main contributions and performances of different state-of-the-art Re-ID approaches on benchmark datasets. This survey also incorporates detailed descriptions of the different benchmark RGB–D, IR, and thermal datasets.

From our rigorous studies on RGB–D and IR sensor-based person re-identification approaches and published datasets, we have observed the following directions in RGB–D and IR sensor-based Re-ID:Available datasets are insufficient for training good models for deep-learning approaches. Though some datasets have a decent number of individuals [58], the number of frames per person is insufficient and is not enough to give the overall variations necessary to build good models.Some RGB–D datasets [14,35,41,58] were collected using a Kinect camera; however, this camera is limited in capturing distant objects because it can only capture objects within 4 m [41]. Therefore, this camera is not suitable for surveillance when individuals exceed a distance of 4.0 m. To overcome this distance limitation, Re-ID researchers can use modern Intel RealSense depth cameras, which can capture images at a range of up to 10 m [100]. Moreover, the dataset [33] did not capture skeleton information properly; even RGB and depth images were not synchronized.A thorough review of state-of-the-art RGB–IR-based cross-modal Re-ID approaches suggests that non-generative models deal with the issue of modality gaps at the feature level while the generative model deals with it at the pixel-level. Although the chances of introducing noise with the generative model are high, it can effectively avoid the effects of color information.

This survey also finds that different approaches relied on different experimental settings, which hinders fair state-of-the-art comparison. Thus, as a future direction, we plan to design a standard experimental set-up and perform experimental evaluations regarding all state-of-the-art approaches to draw a fair comparison.

## Figures and Tables

**Figure 2 sensors-23-01504-f002:**
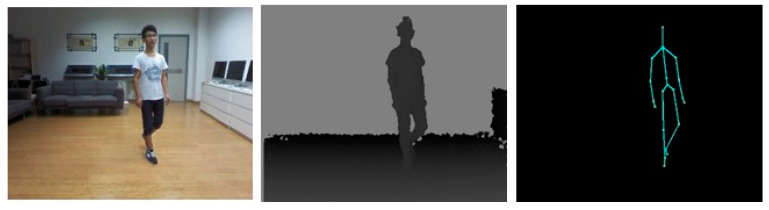
Sample images showing the RGB, depth, and skeleton information of an individual in the RobotPKU RGBD-ID [33] dataset.

**Figure 3 sensors-23-01504-f003:**
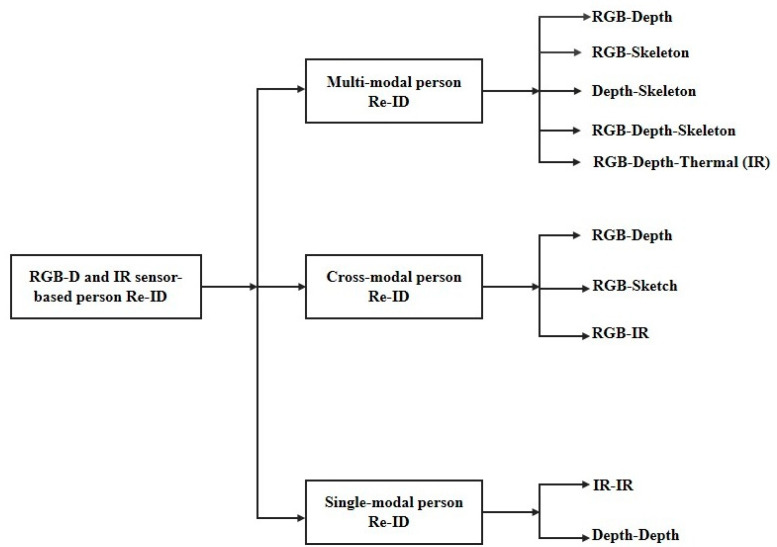
General categories of RGB–D and IR sensor-based person re-identification systems.

**Figure 4 sensors-23-01504-f004:**
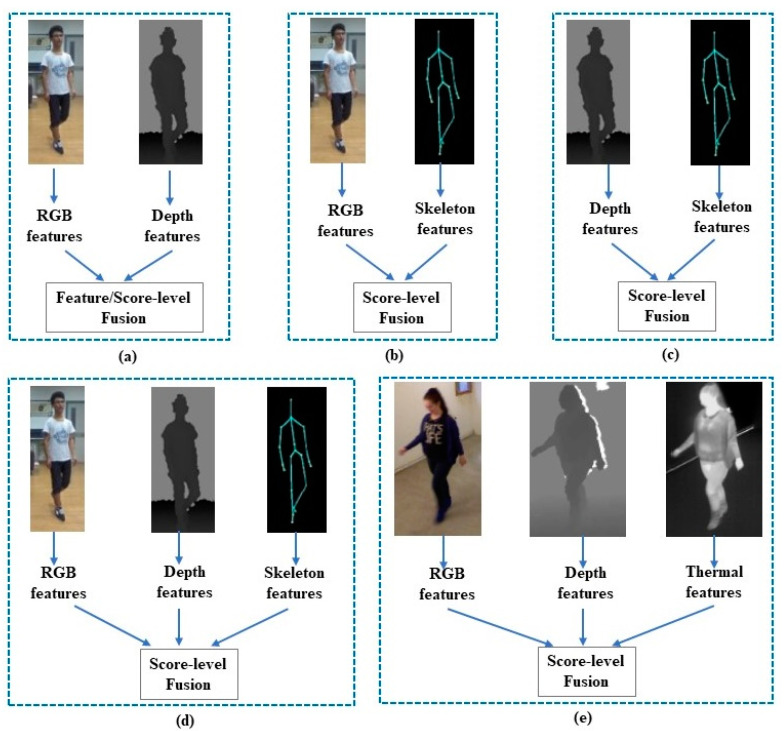
Different multi-modal person re-identification approaches (feature/score-level fusion), where each approach uses different modalities. (**a**) RGB and depth modalities; (**b**) RGB and skeleton information; (**c**) depth and skeleton information; (**d**) RGB, depth, and skeleton information; and (**e**) RGB, depth, and thermal images [44].

**Figure 5 sensors-23-01504-f005:**
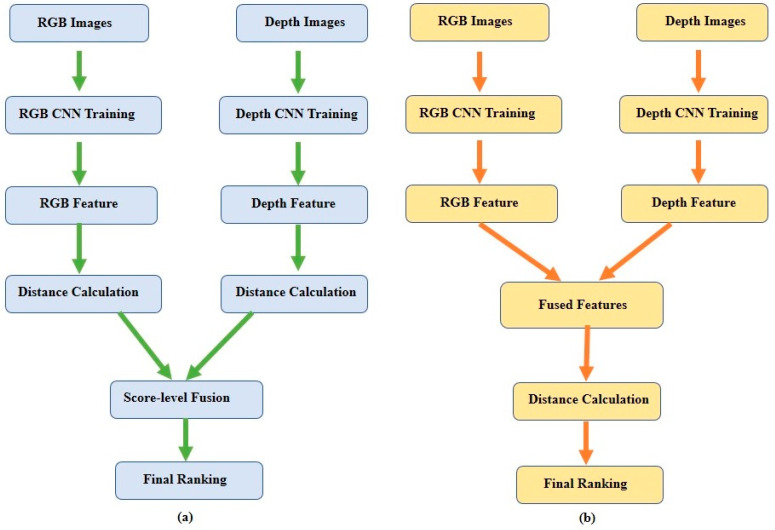
A flowchart of the RGB–depth image-based Re-ID framework. (**a**) Score-level fusion and (**b**) feature-level fusion techniques.

**Figure 6 sensors-23-01504-f006:**
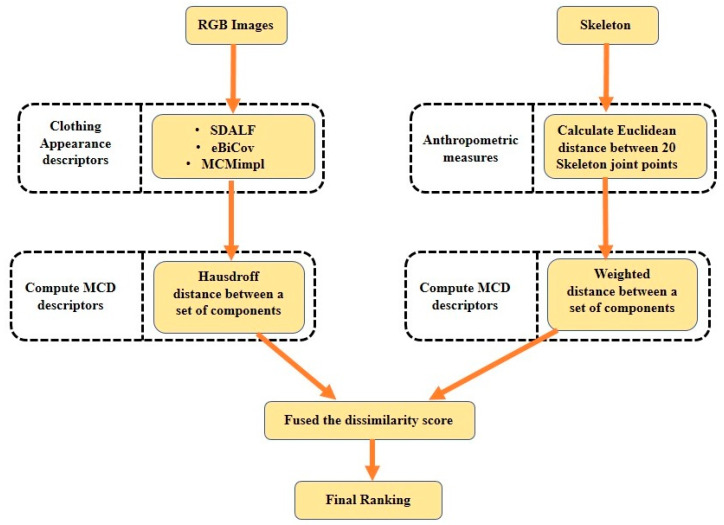
A flowchart of the RGB–skeleton information-based Re-ID framework.

**Figure 7 sensors-23-01504-f007:**
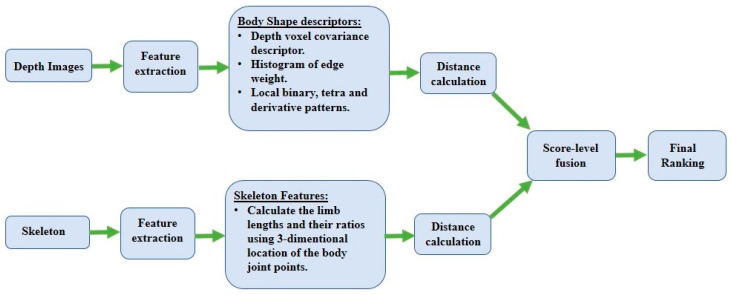
A flowchart of the depth–skeleton information-based Re-ID framework.

**Figure 8 sensors-23-01504-f008:**
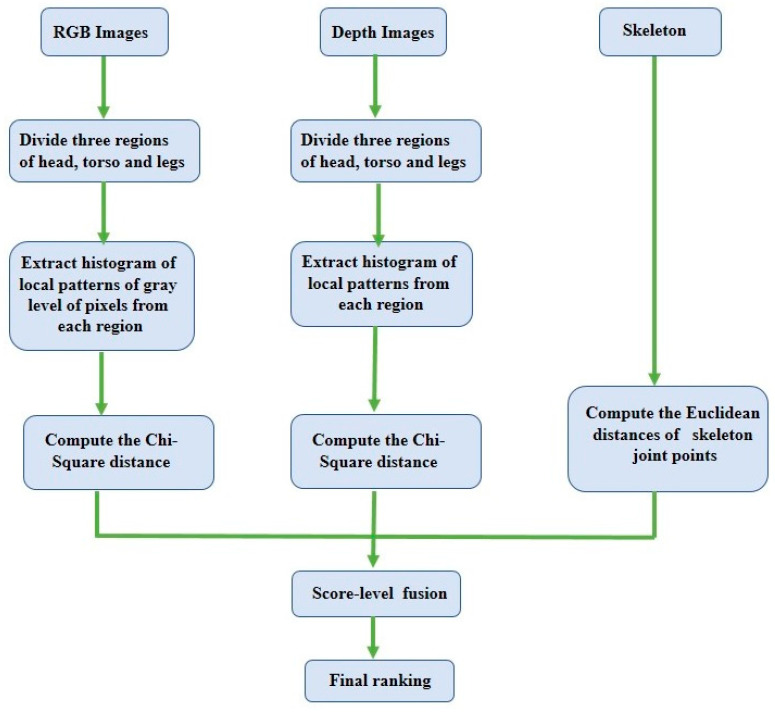
A flowchart of the RGB–depth–skeleton-based Re-ID.

**Figure 9 sensors-23-01504-f009:**
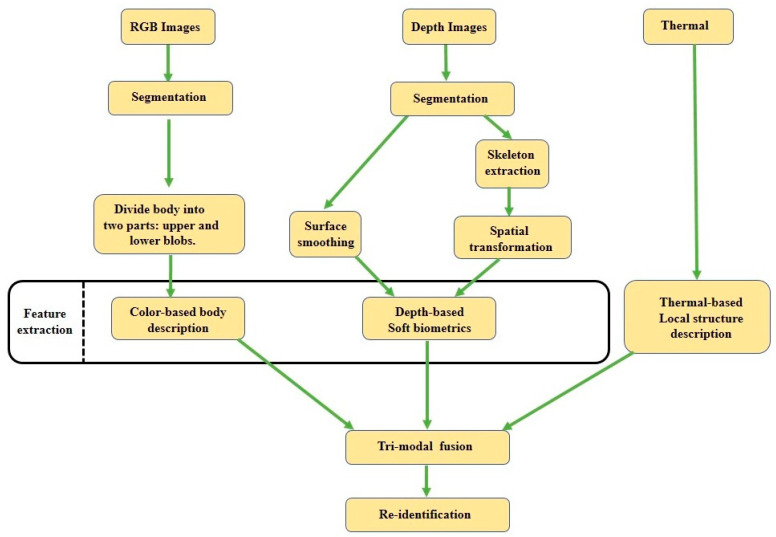
A flowchart of RGB–depth–skeleton-based Re-ID.

**Figure 10 sensors-23-01504-f010:**
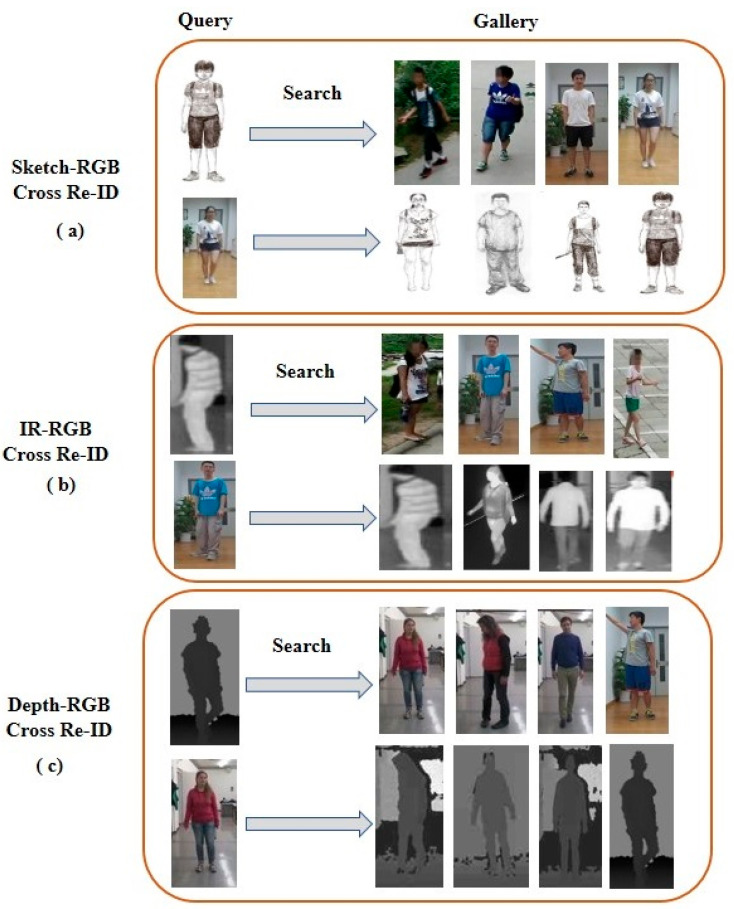
Illustration of cross-modal person Re-ID. (**a**) Sketch–RGB cross-modal Re-ID uses sketch as a query and RGB as a gallery or vice versa. (**b**) IR–RGB cross-modal Re-ID uses IR as a query and RGB as a gallery or vice versa. (**c**) Depth–RGB cross-modal Re-ID uses depth as a query and RGB as a gallery or vice versa.

**Figure 11 sensors-23-01504-f011:**
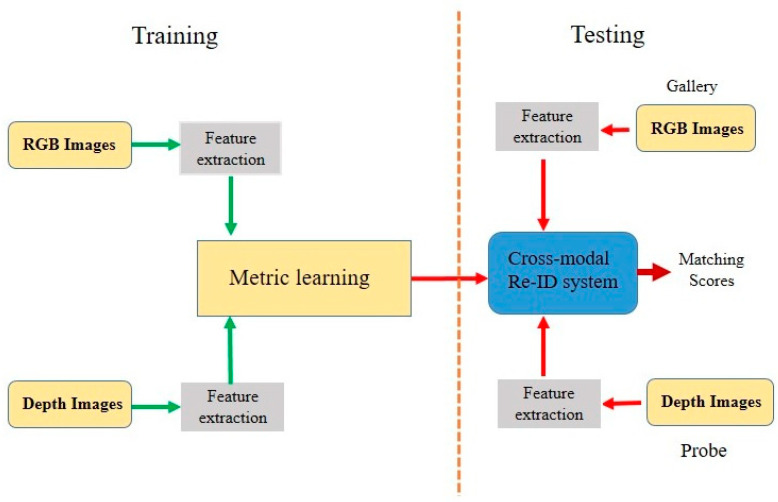
The general architecture of RGB–depth cross-modal Re-ID.

**Figure 12 sensors-23-01504-f012:**
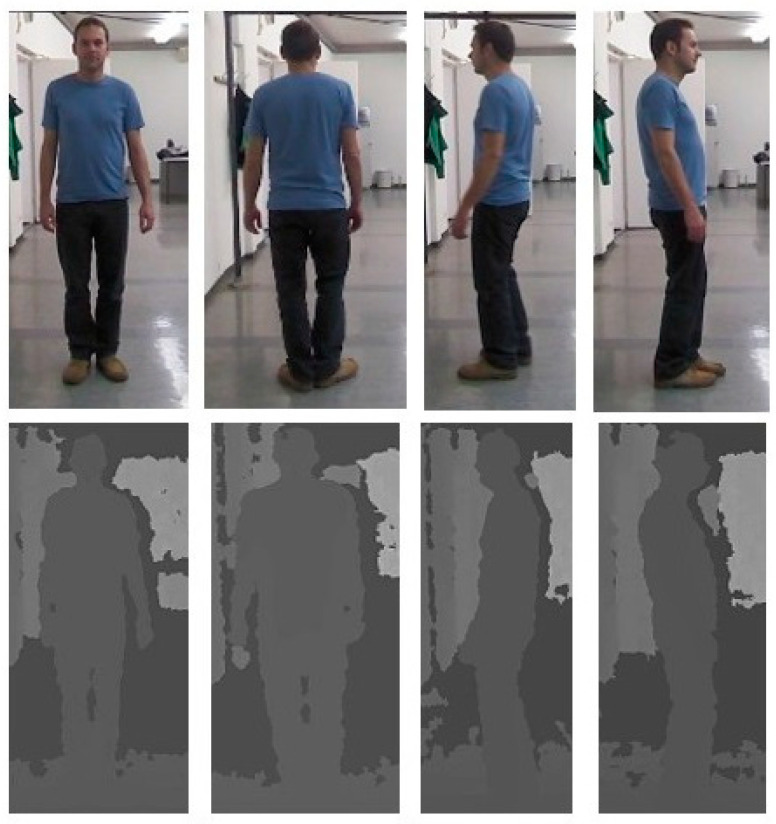
Sample images showing frontal, rear, and side views of an individual, with the camera installed at a horizontal viewpoint.

**Figure 13 sensors-23-01504-f013:**
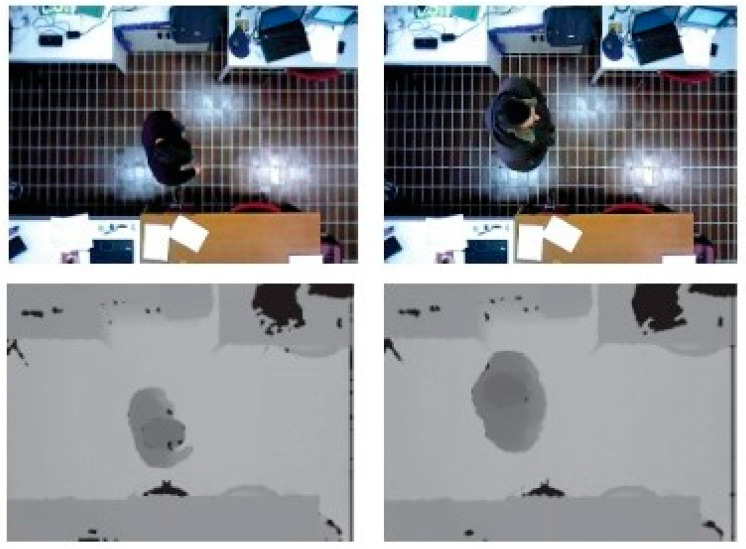
Sample image taken from the TVPR dataset showing an overhead view of a person.

**Figure 14 sensors-23-01504-f014:**
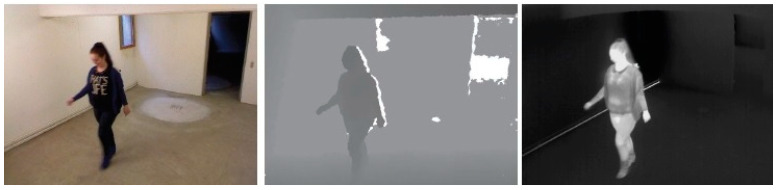
Sample images showing three modalities (RGB, depth, and thermal) taken from the RGB-D-T dataset.

**Figure 15 sensors-23-01504-f015:**
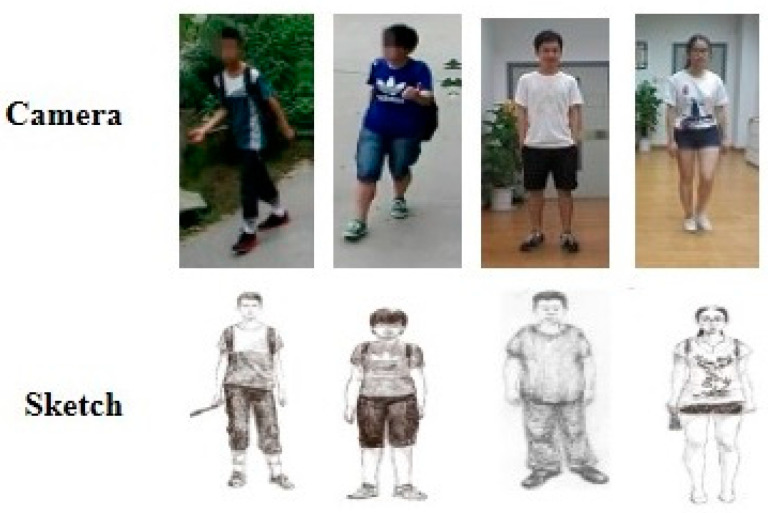
Example images showing sketches of different persons and their corresponding RGB images.

**Figure 16 sensors-23-01504-f016:**
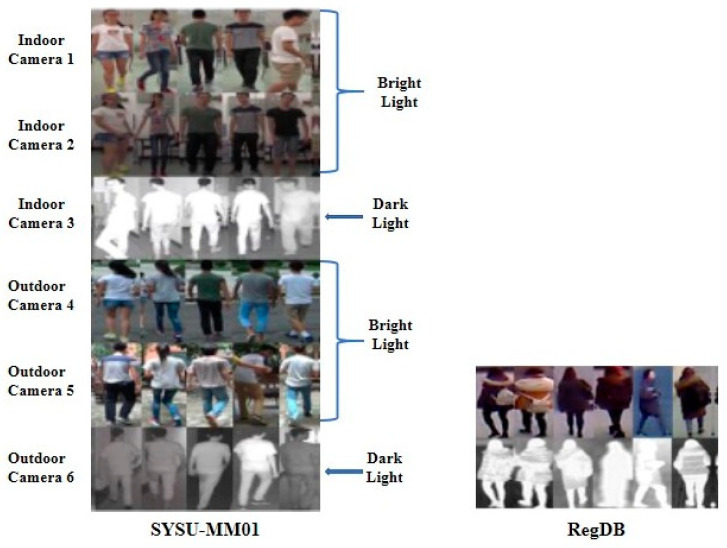
The left figure shows sample images from the SYSU-MM01 dataset and the right one shows sample images from the RegDB dataset.

**Table 1 sensors-23-01504-t001:** Shows summarized information about the different multi-modal state-of-the-art Re-ID approaches.

Reference	Year	Approach	Brief Description	Dataset and Accuracy (Rank 1/nAUC)
Barbosa et al. [58]	2012	Soft-biometric feature-learning approach.	This Re-ID approach has two distinct phases: soft-biometric feature extraction and matching. Soft-biometric features are extracted from depth data, and these features are matched against test samples from the gallery set.	RGBD-ID: 88.1% (nAUC).
Mogelmose et al. [44]	2013	RGB, depth, and thermal feature-based score-level fusion technique.	In this approach, color information from different parts of the body, soft biometrics, and local structural information are extracted from RGB, depth, and thermal data, respectively. This information is combined in a joined classifier to perform the Re-ID task.	RGB–D–T: 82% (rank 1).
Munaro et al. [35]	2014	One-shot person Re-ID with soft-biometric cues.	This approach compares the performance of Re-IDs between skeleton information and point-cloud-estimation techniques. The authors considered four different classifiers, Nearest Neighbor, SVM, Generic SVM, and Naïve Bayes, to compare the performance between them.	BIWI RGBD-ID: For skeleton, 26.6% (rank 1) and 89.7% (nAUC). For point cloud, 22.4% (rank 1) and 81.6% (nAUC).
Munaro et al. [14]	2014	3D-model reconstruction of freely moving people from point clouds which are used to re-identify individuals.	This approach shows how the 3D model of a person is effectively used for person re-identification tasks and also overcomes the issue of pose variation by turning the skeleton information of a person’s point clouds into a standard pose.	BIWI RGBD-ID: 32.5% (rank 1), 89.0% (nAUC). IAS-Lab RGBD-ID: 43.7% (rank 1), 81.7% (nAUC).
Pala et al. [41]	2015	RGB appearance features and skeleton information-based score-level fusion technique.	In this approach, the re-identification accuracy of RGB appearance features is improved by fusing them with anthropometric information extracted from depth data. A dissimilarity-based framework is employed for fusing multi-modal features.	RGBD-ID: 73.85% (rank 1). KinectREID: 50.37% (rank 1).
Imani et al. [36]	2016	Re-identification using local pattern descriptors and anthropometric measures.	In this approach, the histograms of Local Binary Patterns (LBP) and Local Tetra Patterns (LTrP) are computed as features for person Re-ID. These histogram features are fused with anthropometric features using score-level fusion.	RGBD-ID: 76.58% (rank 1). KinectREID: 66.08% (rank 1).
Wu et al. [34]	2017	Depth-shape-based Re-ID approach.	This approach exploits depth–voxel covariance descriptors and local, invariant Eigen-depth features. The authors also extracted skeleton features from joint points of the skeleton. Finally, they combined depth-shape descriptors with skeleton-based features to form complete representations of individuals for re-identification	RGBD-ID: 67.64% (rank 1). BIWI RGBD-ID: 30.52% (rank 1). IAS-Lab RGBD-ID: 58.65% (rank 1).
Ren et al. [37]	2017	Uniform deep-learning-based person Re-ID approach.	This approach uses the deep network to extract anthropometric features from depth images and design a multi-modal fusion layer combining the extracted features and RGB images through the network with a uniform latent variable.	RGBD-ID: 76.7% (rank 1). KinectREID: 97% (rank 1).
Lejbolle et al. [40]	2017	CNN-based multi-modal feature fusion approach.	This Re-ID approach considers an overhead view rather than a frontal view of individuals, which reduces privacy issues and occlusion problems. Two CNN models are trained using RGB and depth images to provide fused features, which improve the accuracy of Re-ID.	OPR: 74.69% (rank 1). TVPR: 77.66% (rank 1). DPI-T: 90.36% (rank 1).
Liu et al. [33]	2017	Person re-identification based on metric model updates.	In this approach, each person is described by RGB appearance cues and geometric features using skeleton information. Then, a metric model is pre-trained offline using label data, and face information is utilized to update the metric model online. Finally, feature similarities are fused using the feature funnel model.	RobotPKU: 77.94% (rank 1). BIWI RGBD-ID: 91.4% (rank 1).
Lejbolle et al. [47]	2018	A multi-modal attention network based on RGB and depth modalities.	This Re-ID approach considers an overhead view for person Re-ID to decrease occlusion problems and increase privacy preservation. A CNN and an attention module are combined to extract local and discriminative features that are fused with globally extracted features. The authors finally fused RGB and depth features to generate joint multilevel RGB–D features.	DPI-T: 90.36% (rank 1). TVPR: 63.83% (rank 1). OPR: 45.63% (rank 1).
Imani et al. [63]	2018	Two novel histogram feature-based Re-ID.	The authors extracted two features, a histogram of the edge weight and a histogram of the node strength, from depth images. Then, the histograms were combined with skeleton features using score-level fusion and were used for person re-identification.	KinectREID: 58.35% (rank 1). RGBD-ID: 62.43% (rank 1).
Patruno et al. [59]	2019	Person Re-ID using Skeleton Standard Postures and color descriptors.	This approach introduces Skeleton Standard Postures (SSPs) for computing partition grids to generate independent and robust color-based features. A combination of the color and the depth of information identifies very informative features of a person to increase re-identification performance.	BIWI RGBD-ID: 97.84% (rank 1). KinectREID: 61.97% (rank 1). RGBD-ID: 89.71% (rank 1).
Ren et al. [38]	2019	Uniform and variational deep learning for person Re-ID.	This approach uses uniform and variational deep learning for person re-identification. The authors extracted RGB appearance features and depth features using one CNN for each feature from RGB–D images. To combine the appearance features and depth features, uniform and variational auto-encoders were designed on the top layer of the deep network to find a uniform latent variable.	KinectREID: 99.4% (rank 1). RGBD-ID: 76.7% (rank 1).
Imani et al. [55]	2020	Re-identification using RGB, depth, and skeleton information.	In this approach, the depth and RGB images are divided into three regions: the head, the torso, and the legs. For each region, a histogram of local vector patterns is estimated. The skeleton features are extracted by calculating the various Euclidean distances for the joint points of skeleton images. Then, extracted features are combined as double and triple combinations using score-level fusion.	KinectREID: 75.83% (rank 1). RGBD-ID: 85.5% (rank 1).
Uddin et al. [48]	2020	Depth-guided attention of person Re-ID.	This approach introduces depth-guided, attention-based person re-identification. The key component of this framework is the depth-guided foreground extraction that helps the model improve the performance of Re-ID.	RobotPKU: 92.04% (rank 1).
Martini et al. [57]	2020	A deep-learning approach for top-view, open-world person Re-ID.	The authors present top-view, open-world person Re-ID. This approach is based on a pretrained deep CNN, finetuned using a dataset acquired by using top-view configuration. A triplet loss function is used to train the network.	TVPR: 95.13% (rank 1). TVPR2 [42]: 93.30% (rank 1).
Uddin et al. [42]	2021	Fusion in a dissimilarity space for RGB–D person Re-ID.	In this approach, two CNNs are separately trained with three-channel RGB and four-channel RGB–D images to produce two different feature embeddings and compute the dissimilarities between queries and galleries in different feature embeddings. The computed dissimilarities for two individual modes are then fused in a dissimilarity space to obtain their final matching scores.	RobotPKU: 93.33% (rank 1). RGBD-ID: 82.05% (rank 1). SUCVL: 87.65% (rank 1).

**Table 2 sensors-23-01504-t002:** Shows summarized information about the multi-modal datasets. ‘*’ indicates the dataset is not publicly available.

Dataset	Year	#ID	#Camera and Location	RGB	Depth	Skeleton
RGBD-ID	2012	79	1 (same location but different clothes and poses)	√	√	√
BIWI-RGBD-ID	2014	50	1 (same location but different days with different clothes and poses)	√	√	√
IAS-Lab	2014	11	2 (different locations and people wearing different clothes)	√	√	√
KinectREID	2015	71	1 (same room with different poses)	√	√	√
RobotPKU RGBD-ID	2017	90	2 (two different rooms with pose variations)	√	√	√
SUCVL RGBD-ID *	2021	58	3 (three different indoor locations)	√	√	

## Data Availability

Not applicable.

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
