# Peer review of "Person Re-Identification with RGB–D and RGB–IR Sensors: A Comprehensive Survey"

_sensors, 2023, doi:10.3390/s23031504_

Round 1

Reviewer 1 Report

This is an overview article on person re-identification with RGB-D and RGB-IR Sensors. The author lists state-of-the-art methods on person re-identification using RGB-D and RGB-IR sensors and classifies them as well as analyzes the experimental results. The language of the paper is fluent, the logic is clear, and the data is reliable, which has certain reference value and significance. However, the paper needs to be improved in the following three aspects:

1. As an overview paper, each type of algorithm mentioned in the paper should be analyzed more deeply, the differences, advantages and disadvantages, and scope of application of each type of algorithm mentioned in the paper need to be discussed more thoroughly.

2.  When introducing algorithms, it is better to add flow charts or pseudo codes to make readers understand the characteristics of different algorithms more clearly.

3. The logical relationship between section 5 and the first four sections has not been fully elaborated, and it is suggested to add a transition part to further sort out the context of the article.

Reviewer 2 Report

In this paper, the authors conducted a comprehensive review of existing Re-ID approaches which utilize the different sensor-based additional information to address those constraints that face RGB camera-based person Re-ID systems. The entire survey is written well however, I have major comments about the current version.

·       The overall writing is very weak, I recommend the authors deeply proofread the manuscript from a native English team.

·       In the introduction section, some latest works are missing which purely related to this work.

https://doi.org/10.3390/electronics11030454 ; DOI: 10.1002/int.22820 ; DOI10.1109/TIFS.2022.3218449

·       Arrange all the figures in a proper format, also in the methodology, provide a figure for each subsection for the ease of reader understanding.

·       Better to make one table in section-2, and mention the paper's technique from (2010-2022), a short description, and a dataset covered in this study.

Round 2

Reviewer 1 Report

This paper reveals a detailed taxonomy of existing methods and existing RGB-D and RGB-IR person re-ID datasets, and summarizes the performance of state-of-the-art methods on several representative RGB-D and RGB-IR datasets. The revised paper summarizes comprehensively, in-depth, and in place, and the data sources are true and reliable. At the same time, the paper also introduces the advantages and disadvantages of each method, the suitable angle and scope of application, and analyzes the characteristics of various algorithms from the aspects of development prospects, practical applications, and algorithm complexity. The whole paper is clear and has certain reference value and significance. Therefore, it is recommended to accept.
